# Dysbiosis and Alzheimer’s Disease: A Role for Chronic Stress?

**DOI:** 10.3390/biom11050678

**Published:** 2021-04-30

**Authors:** Vivek Kumar Sharma, Thakur Gurjeet Singh, Nikhil Garg, Sonia Dhiman, Saurabh Gupta, Md. Habibur Rahman, Agnieszka Najda, Magdalena Walasek-Janusz, Mohamed Kamel, Ghadeer M. Albadrani, Muhammad Furqan Akhtar, Ammara Saleem, Ahmed E. Altyar, Mohamed M. Abdel-Daim

**Affiliations:** 1Chitkara College of Pharmacy, Chitkara University, Punjab 140401, India; v4vksharma@gmail.com (V.K.S.); nikhilgarg096@gmail.com (N.G.); sonia.dhiman@chitkara.edu.in (S.D.); saurabh.gupta@chitkara.edu.in (S.G.); 2Goverment College of Pharmacy, District Shimla, Rohru 171207, India; 3Department of Pharmacy, Southeast University, Banani, Dhaka 1213, Bangladesh; pharmacisthabib@gmail.com; 4Laboratory of Quality of Vegetables and Medicinal Plants, Department of Vegetable Crops and Medicinal Plants, University of Life Sciences in Lublin, 15 Akademicka Street, 20-950 Lublin, Poland; agnieszka.najda@up.lublin.pl (A.N.); magdalena.walasek@up.lublin.pl (M.W.-J.); 5Department of Medicine and Infectious Diseases, Faculty of Veterinary Medicine, Cairo University, Giza 12211, Egypt; m_salah@cu.edu.eg; 6Department of Biology, College of Science, Princess Nourah bint Abdulrahman University, Riyadh 11474, Saudi Arabia; gmalbadrani@pnu.edu.sa; 7Riphah Institute of Pharmaceutical Sciences, Riphah International University, Lahore Campus, Lahore 54950, Pakistan; furqan.pharmacist@gmail.com; 8Department of Pharmacology, Faculty of Pharmaceutical Sciences, Government College University Faisalabad, Faisalabad 38000, Pakistan; amarafurqan786@hotmail.com; 9Department of Pharmacy Practice, Faculty of Pharmacy, King Abdulaziz University, P.O. Box 80260, Jeddah 21589, Saudi Arabia; aealtyar@kau.edu.sa; 10Pharmacology Department, Faculty of Veterinary Medicine, Suez Canal University, Ismailia 41522, Egypt

**Keywords:** Alzheimer’s disease, microbiota, probiotics, chronic stress, glucocorticoids, HPA Axis

## Abstract

Alzheimer’s disease (AD) is an incurable, neuropsychiatric, pathological condition that deteriorates the worth of geriatric lives. AD is characterized by aggregated senile amyloid plaques, neurofibrillary tangles, neuronal loss, gliosis, oxidative stress, neurotransmitter dysfunction, and bioenergetic deficits. The changes in GIT composition and harmony have been recognized as a decisive and interesting player in neuronal pathologies including AD. Microbiota control and influence the oxidoreductase status, inflammation, immune system, and the endocrine system through which it may have an impact on the cognitive domain. The altered and malfunctioned state of microbiota is associated with minor infections to complicated illnesses that include psychosis and neurodegeneration, and several studies show that microbiota regulates neuronal plasticity and neuronal development. The altered state of microbiota (dysbiosis) may affect behavior, stress response, and cognitive functions. Chronic stress-mediated pathological progression also has a well-defined role that intermingles at various physiological levels and directly impacts the pathological advancement of AD. Chronic stress-modulated alterations affect the well-established pathological markers of AD but also affect the gut–brain axis through the mediation of various downstream signaling mechanisms that modulate the microbial commensals of GIT. The extensive literature reports that chronic stressors affect the composition, metabolic activities, and physiological role of microbiota in various capacities. The present manuscript aims to elucidate mechanistic pathways through which stress induces dysbiosis, which in turn escalates the neuropathological cascade of AD. The stress–dysbiosis axis appears a feasible zone of work in the direction of treatment of AD.

## 1. Introduction

Increased age, owing to medical advancement, has made humans prone to a variety of complications and illnesses. Dementia is such an age-associated neurodegenerative and neuropsychiatric complication that affects, devastates, and deteriorates geriatric lives. Alzheimer’s disease (AD) is the most common form of dementia that has shown an upsurge worldwide. There are approximately 66 million people suffering from AD and this number is unfortunately predicted to increase up to 115 million by 2030 [1]. Between 2000 and 2017, the death rate showed a 31% rise in the age group of 65 to 74, an upsurge of 57% for people aged 75 to 84, and 86% for people aged 85 and older [2]. The associated health care costs are also expected to increase to 1 trillion from the existing $305 billion in 2020 in the coming years. This associated health care cost includes expert nursing care, hospital care, and home healthcare, and the costs include informal care giving that is likely underestimated and are associated with significant negative societal and personal burden [3]. Thus, rather than a neurodegenerative sickness, AD has become a social burden that requires immediate, expert, timely, and result-oriented intervention. Typically, AD is exemplified by impairments in episodic memory followed by deterioration of the cognitive domain that includes malfunctioned state of visuospatial, linguistic, behavioral, and other executive functions [4]. The major pathological markers of AD (hippocampal atrophy, amyloid plaques, neurofibrillary tangles, neuronal death) can be identified years before the cognitive decline and neuropsychiatric disturbances [5,6]. Oxidative stress, increased age, genetics, neuroinflammation, etc. have found a well-established place among etiopathology of AD; furthermore, the interaction of genetic and several environmental risk factors are found to be associated with AD. Among them, alteration in the gut microbiota is now recognized as a viable target to be investigated [7]. The microbiota has shown close involvement with several neurodegenerative and neuropsychiatric complications including AD, amyotrophic lateral sclerosis (ALS), Parkinson’s disease (PD), and Huntington’s disease (HD) [8]. The benefit and use of microbes have been advocated by several studies which report the significant cognitive deficits and neuronal deterioration in their altered state. Thus, the absence of microbes is a corroborating critical link between the gut–brain axis and cognition. The microbiota domain is not a new area to target neurodegeneration but has been tried since 1920, when probiotics containing lactobacillus were used to improve mental health and psychiatric disorders. This approach was not much appreciated and accepted due to the non-revelation of their clear-cut mechanisms. Recent times have witnessed a resurge in the interest and role of gut microbes in the improvement of mental health, owing to their ability to improve psychiatric wellbeing through their interaction with different physiological systems [9]. The interest in the microbial domain grows further as the microbiota of AD patient’s shows huge variations and differences in taxonomy and phyla with the control and there is a decrease in Firmicutes, an increase in Bacteroidetes, and a decrease in Bifidobacterium compared to healthy controls [10]. This state of dysbiosis is complemented by dysregulation of the anti-inflammatory P-glycoprotein pathway and signaling of pro-inflammatory mechanisms (altered epithelial permeability, decrease mucus production, and enhanced passage of pro-inflammatory cytokines into the systemic circulation) [11]. Some studies have tied cognitive decline and AD risk with exposure to common pathogens including Cytomegalovirus, Herpes simplex virus type 1, Helicobacter pylori, etc. which have shown abundance in AD patients [12]. Additionally, gut microbes influence brain attributes through various ways, which include dysbiosis-related inflammation, the involvement of altered microbiota in oxidative stress, altered permeability, resultant immune activity, age-related reduction in beneficial microbiota, dysbiosis induced insulin-resistant state, etc. These are the major mechanistic and pathological links through which cognitive domain and pathological progression of AD are influenced [13]. The microbial connection was earlier thought to be limited to the processes involved in the fermentation of sugar, synthesis of vitamins, and metabolism of xenobiotics only, but their role in various disorders, including diabetes, cardiovascular complications, and obesity, advocates their role in AD, as most of these metabolic complications are also associated with the occurrence of AD. These findings are supported by differences in the quantitative and quantitative differences of microbiota between normal individuals and AD sufferers [14,15]. There are approximately more than 1000 species and 7000 strains of bacteria that reside in the human gut. The human gastrointestinal tract (GIT) is the largest and richest reservoir of microorganisms in the human body. Most of them (approximately 99%) are anaerobic bacteria, while aerobic genera (Enterococcus and Enterobacteriaceae) constitute the sub-dominant portion [16]. Fungi, virus, yeast, archaea, and protozoa are also present in minute quantities (approximately 1%) [17]. In total, 95% of symbiotic microorganisms are present in the gut, and this microbiome has a diverse role in human physiology. Gram-positive bacteria are predominantly present in the oral and periodontic region and gram-negative dominate in the stomach and intestinal areas. Microorganisms populating the gut (microbiota) and human healthiness have an important and interesting relationship because not only symbionts and commensals, but also pathogenic microorganisms constitute the microbiota [18]. Thus, a well-balanced state of microbiota is indispensable for homeostasis owing to their role in the regulation of body weight, glucose metabolism, neurotropism, immunomodulation, hypersensitivity, inflammation, and overall, in the regulation of normal growth and development [19]. The difference in bacteria composition determines the difference in the biochemistry of individuals, which is responsible for genetic susceptibility and resistance to diseases [20]. Firmicutes (51%), Bacteroides (48%), Proteobacteria, Actinobacteria (Bifidobacterium), and Cyanobacteria are the major bacterial phyla constituting GIT and share a mutual positive association with the host [13]. Owing to the diversity of microbiota, it regulates multiple neuro-chemical, neuro-immune, and neuro-metabolic pathways through a highly interactive host–microbiome signaling that links various organs with major parts of the brain. Gut microbiota controls several neurochemical pathways, including the hypothalamic–pituitary–adrenal (HPA) axis, the central stress response system, and immunogenic mechanisms, through which they are assumed to influence cognitive processes through interaction with the gut–brain axis. The benefits of microbiota have been also proven by the benefits of leptin, ghrelin, GLP-1, and GIP, which further gives an idea of how those peptide hormones in GIT influence neurological functions and advocate microbial metabolic deformities in neurodegeneration. Additionally, the use of probiotics (*lactobacillus* and *bifidobacteria*) in AD reversed various pathological markers, including atrophy and amyloid accumulation, and influenced proteasome degradation of Amyloid β and autophagy. The role of microbiota has also been documented in various neuropsychiatric illnesses and stress-mediated disorders, which reflects the microbiota as an ever-changing and dynamic domain that can be targeted for neurodegenerative complications including AD [21].

Stress threatens homeostasis, challenges the wellbeing of an individual, and impairs the physiological balance of an individual. Stress involves activation of the HPA axis and release of glucocorticoids. Stress mechanism activates apoptotic pathways, aggravates oxidative stress, and causes inflammation. All these factors are major etiological influences for various neurodegenerative disorders and may predispose the nervous system to the subsequent development of neuro-damage during the lifetime. In relation to AD, the effects of stress on the hippocampus are of prime consequence, where neuroinflammation and apoptosis in response to stress damages neurons [22]. Peripheral inflammation of GIT due to microbiota modulation or inflammation and oxidative stress also influence prime CNS modulators and affect behavior and cognition. The intestinal manifestations influence progression of neuropsychiatric illnesses through diverse mechanisms that include alteration of intestinal permeability, dysregulation of serotonin levels, activation of mast cells, and affecting neurotrophic factors. In many studies, it has been reported that peripheral inflammation can disturb the blood–brain barrier integrity and allows the spread of peripheral inflammation to CNS. It has been noticed that the level of tight junctional proteins (occluding and Claudin-5) was downregulated, while levels of caspase-3 were upregulated in cortex and hippocampus. These observations points toward the impact of peripheral inflammation- and oxidative stress-induced apoptotic cascade in brain regions that may be responsible for neuronal demise [23].

## 2. Microbiota-Physiological Role: Gut Brain Axis

There are certain limitations to grow and culture intestinal microbes; thus, while their exact number has not been defined, they are assumed to have up to 1012 per/mL density, which is the highest in any microbial ecosystem. The human microbiota encodes almost 4 × 106 genes, which is a hundred times more than the number of human genes [24,25]. The microbiota provides the widest and vastest interface between an individual and the outside world, with an estimated area equivalent to a tennis court (260–300 m). This community of microbiota outnumbers the number of eukaryotic cells in the human body [26]. From a historical point of view, William James and Carl Lange conceptualized that bidirectional communication between the GIT and the brain may influence cognitive and emotional attributes, while Walter Cannon proposed that neuronal signals steer GIT functions. Now, the gut–brain axis is an emerging field in pharmacology, gastroenterology, microbiology, psychology, and translational research [27,28]. The gut–brain axis requires optimal functioning of the endocrinal, sympathetic–parasympathetic, and enteric nervous system and a state of dysbiosis may prove detrimental for them. The fetal gut is almost sterile and microbial colonization starts once the fetus passes from the uterus. The microbial composition is influenced by maternal nutrition and the mode of delivery [29]. Infant microbiota has a maternal influence and is mainly inhabited with lactobacillus and prevotella species, and those born through cesarean section have Staphylococcus and Propionibacterium as the dominant species. Breast-fed infants also have a high concentration of bifidobacteria, while this number is much lower in infants fed with artificial food. The initial colonization of microbiota is particularly important and decisive for permanent composition; therefore, environmental and nutritional stressors have an important role to shape the microbiota [30]. This community diversifies and expands over time and is mainly dominated by anaerobes in adults [31] and microbiota show a near-final shape by three years of age, with a shift from Bifidobacterium to Clostridia and Bacteriodetes [32]. The composition of microbiota in the human system is also influenced by the anatomical site, pH, antibiotic use, drugs, stress, moisture, dietary habits, etc. [33]. The host system provides a nourishing and beneficial environment to microbes, and in turn, they metabolize undigested food, including xenobiotics, and provide protection against pathogens through their immunomodulatory and anti-inflammatory action [34]. The commensal or beneficial microbes also help in the differentiation and growth of epithelial barriers providing protection against pathogens. Counting for their immuno-modulatory functions, microbiota expands the gut-associated and intraepithelial lymphoid tissue and fine-tunes the T helper cells and cytokine differentiation [35]. Microbiota aids in the digestion of foods, colonization resistance (bacterial antagonism through the production of antimicrobial substances), and mucus secretory functions. Microbiota also induces the expression of genes in enterocytes that gives an added advantage to the residing bacteria and makes it difficult for pathogens to get attached and colonize [36]. The microbial metabolites are utilized to produce amino acids (tryptophan, GABA) and monoamines (serotonin, dopamine, etc.), which reach the neuronal entity through the bloodstream and have a pivotal role in neurotransmission and cognition [37] (Figure 1). The microbiota has a multitude of functionality in intestinal barrier homeostasis, bone homeostasis, acute stress responsiveness, and neurogenesis as well as in the regulation of behavior and mood [38]. Their role in the fermentation of indigestible residues, digestion of indigestible carbohydrates, vitamin synthesis (especially B group vitamins serving as substrates/cofactor in several biochemical reactions), and in the metabolic recovery of energy to complete biochemical pathways [16] makes them indispensable for survival. A more diverse and healthier microbiota not only regulates crucial processes of digestion and aids extraction and absorption of metabolites, but also maintains intestinal epithelium integrity by secretion of bacteriocins. Microbiota also ensures the availability of SCFA (butyrate, propionate, and acetate), which are also indispensable for mucin synthesis and anti-inflammatory activities, and ensures gut permeability. An altered microbiota leads to altered membrane permeability, which propagates inflammation and alters normal signaling between the neural–endocrine–HPA axis, which ultimately affects brain functions [39,40]. The bidirectional communication between the brain and enteric nervous system in the abdomen transpires through the modulation of hormones and signaling of sympathetic and parasympathetic nervous systems networks. These complicated yet closely divulged interactions form the basis of linking two systems together. This axis now includes microbiota as functions of the gut (ENS), which are manipulated to a large extent by the microbiome. This is logical, as the microbiome influence ENS, which in turn has potential effects on the brain and microbiota and has emerged as a viable area to diagnose and therapeutically target for neurological and neurodegenerative disorders [41]. Microbiota can influence the ENS, and both the CNS and ENS share several similarities morphologically, physiologically, and pharmacologically; thus, if a microbiota has an influence on the ENS, it can also impact the CNS in diverse ways [42].

## 3. Microbiota in AD

Alteration in the microbiota forms the basis for multiple disorders, including asthma, hypersensitivity reactions, autism, multiple sclerosis, diabetes, colorectal cancer, inflammatory bowel disease, obesity, cardiovascular complications (hypertension), neurodegenerative diseases (AD, PD), and neuropsychiatric complications (anxiety, depression, etc.) [43,44]. Chronic antibiotic use, acute/chronic infections, stress, genetics, dietary changes, and increased age are the major factors responsible for altered microbiota (dysbiosis) [45,46]. With the number of unsuccessful efforts to target AD, novel areas, such as the microbiota and the use of probiotics, have gained considerable attention over the years. The vagus nerve serves as the mode of communication between the ENS/gut and autonomic nervous system, and it ends at the brain stem nuclei, which receive and forward afferent and efferent fibers. Thus, the brain stems influence gut functions and propagates signals to the thalamus and cortex. Microbiota form a communication channel between the ENS and CNS, and any change in intestinal permeability leads to infiltration of cytokines and immune cells which influence both the ENS and CNS and forms a crucial modulator of the gut–brain axis [47]. Dysbiosis is a state of quantitative and qualitative changes in a microbiota, which then turns hostile to the host. This is characterized by the unequivocal distribution of microorganisms, overpowering of beneficial effects of commensals/symbionts by pathogens, and resultant changes in their metabolic activity [48]. The state of dysbiosis involves fewer beneficial microbiota and less diverse microbiota, decreased integrity of epithelial barrier, altered membrane permeability, and decreased mucosal defense, leading to the occurrence of infections and serious illnesses. The main genera of beneficial bacteria include Lactobacillus and Bifidobacterium (favorable modulation of GIT physiology, aids in digestion, prevents pathogenic colonization) and *Eubacterium, Roseburia*, and *Faecali bacterium* (antioxidant, anti-inflammatory, maintenance of gut permeability), which possesses favorable functions and maintain homeostasis. The genera Enterobacteriaceae, a family including the gut commensals *Escherichia*, *Shigella*, *Proteus*, and *Klebsiella*, is considered mainly harmful, but the distinction of beneficial or harmful bacteria is not clear, as the dominance of one taxon is not desirable and beneficial bacteria may prove harmful if they enter systemic circulation [49]. The state of dysbiosis is characterized by increased inflammation, as there is a decrease in the anti-inflammatory bacterial population which may signal the response of inflammatory cytokines (IL-1, TNF-α) [50]. Metabolic deformities, such as insulin resistance, dyslipidemia, oxidative stress, hyperglycemia, and alteration in micronutrients, also find an adjacent connotation with AD pathogenesis and are regarded as probable etiological factors for AD, which find a close association with the microbiota [45]. Microbiota can modify behavioral and electrophysiological properties of the nervous system and has also found an association with the cognitive domain of an individual. These findings are supported by the benefits of probiotics in cognitive and behavioral deficits, although the supporting data need to be strengthened further [51,52]. The microbiota has emerged as an important player in synaptic transmission and the formation of neurotransmitters (acetylcholine, dopamine, and serotonin). The animals with no microbiota (germ-free) have shown declined levels of BDNF, NMDA, and GABA, which are implicitly involved in cognitive functions [53]. These germ-free animals also showed deficits in spatial and working memory, suggesting a role of microbiota in the regulation of cognitive functions. All the neurotransmitters and trophic factors produced by microbiota are involved in synaptogenesis, synaptic signaling, excitatory/inhibitory neurotransmission, and the constitution of normal behavior. Thus, by influencing neurotransmitter signaling, receptor expression, and neuromodulatory activities, microbiota may influence cognitive functions closely [54]. The infection of H. pylori in AD patients signals inflammation, the formation of tangles, and cognitive decline, and increased deposition of senile plaques has been reported in the presence of *Orrelia burgdorferi* and *Chlamydia pneumonia* beside pylori infection [12,55]. Blood analyses of AD patients having amyloid load and inflammatory markers have also shown an increased presence of bacteria which instigate inflammation (*Escherichia/Shighella*) and an abridged existence of anti-inflammatory (*Escherichia rectale*) microbes [56]. The germ-free studies (rearing animals in a sterile environment and gnotobiotic units that prevent bacterial colonization) have revealed decreased mRNA BDNF expression (in a sex-dependent manner), abnormal innate immune response, microglial activation, and decreased expression for NMDA receptors [57]. Germ-free mice have also depicted altered microglia numbers, maturation patterns, and immune responses [58]. In various experimental studies, the beneficial effects of the rational composition of microbiota through the administration of probiotics have been documented. The treatment of Bifidobacteria (*B. longum* and *B. breve*) for 11 weeks in BALB/c mice had pro-cognitive effects and, in these animals, learning, memory, and recognition were improved in a strain-specific manner [59]. Additionally, probiotics, when served with fermented milk, not only changed the intrinsic activity of the resting brain but also modified the emotions and sensation [60]. *L. acidophilus, L. fermentum, Bifidobacterium lactis*, and *B. longum*, when given in a combined formulation, decreased oxidative stress and improved memory deficits by modulating microbiota in Aβ injected rats [61]. Furthermore, a combination preparation of *Bifidobacterium lactis, Lactobacillus casei, Bifidobacterium bifidum,* and *Lactobacillus acidophilus* called Probiotic-4, when administered to 9-month-old senescence-accelerated mouse prone 8 (SAMP8) mice for 12 weeks, decreased memory deficits, cerebral neuronal and synaptic injuries, and glial activation and modulated the microbiota. This formulation also lessened the age-associated disruption of the intestinal barrier and blood–brain barrier, decreased LPS, TLR expression, proinflammatory cytokines (TNF-α, IL-6) at both mRNA and protein levels, and nuclear factor-kB (NF-kB) translocation in the brain [62]. Thus, probiotics in general may have an effect on memory deficits and other disorders associated with distorted neuronal functions (Figure 2).

### 3.1. Microbiota, Neurotransmitters, and Trophic Factors

Neurotransmitters (acetylcholine, GABA) and trophic factors have a crucial role in AD pathophysiology and are involved in neuronal communication, synaptic plasticity, and signaling pathways. Both acetylcholine and GABA are involved in cognitive functions and the only approved treatment options for AD also includes the restoration of cholinergic functions through acetylcholinesterase inhibitors (Donepezil, rivastigmine, etc.). Trophic factor withdrawal or deficiency of BDNF also affects synaptic plasticity and escalates neurodegeneration [63]. Lactobacilli and Bifidobacteria are the major microbial component of the microbiome, which converts monosodium glutamate into GABA, increasing its availability in CNS, leading to the upregulation of inhibitory signaling through GABAergic neurons [64]. In a state of dysbiosis, when the proportion of these two strains is downregulated and the intestinal production of GABA is affected, there is the emergence of the state of imbalance of GABA/glutamate activity, which increases the risk of excitotoxic cell death [65]. Additionally, the downregulation of NMDA mRNA expression in the hippocampus in germ-free mice suggests an increase in glutamatergic activity and a relation between gut metabolism and glutamatergic neurotransmission through microbiota [66]. The concentration of glutamate is also somewhere dependent on the microbiota. Glutamate is a major excitatory neurotransmitter that acts on the N-methyl-D-aspartate (NMDA) receptor; glutamate is a prerequisite for excitatory neurotransmission and neuronal survival, neuronal differentiation, synaptic signaling, and balanced neural transmission. This NMDA receptor-mediated excitatory signaling is important for learning and memory. In germ-free mice, the decreased hippocampal mRNA expression of NMDA and antibiotic treatment resulted in decreased NMDA expression shows the influence of microbiota on excitatory neurotransmission in CNS [55,65]. Serotonin has a decisive role to play in the regulation of diverse components of memory through its interaction with glutamatergic, cholinergic, and neuro-modulatory activities. Serotonin, which is mainly tied to memory and cognition, also controls behavioral and psychological symptoms of AD (anxiety, depression, hallucinations, delusions, etc.). Although the pancreas, bones, and mammary glands are entrusted with the production of serotonin, more than 90% of serotonin production occurs through GIT microbiota. The essential amino acid tryptophan is the major substrate for serotonin synthesis. A major part of tryptophan is required for kynurenine synthesis, leading to the generation of nicotinamide adenine dinucleotide (NAD) in humans as well as in the microbiome. Most of the serotonin synthesis involves enterochromaffin cells in the GIT epithelium, which account for 90% of serotonin synthesis. This involves a precise balance of utilization of bacteria and epithelial uptake of tryptophan for serotonin synthesis. The gut microbiota diverts its uptake to the bacterial kynurenine pathway to the detriment of serotonin synthesis by epithelial cells. The metabolites of tryptophan metabolism include kynurenine and quinoline, which have been proposed to perturb brain functions [67]. Thus, the bacterial populations, including Escherichia coli and Enterococci, regulate serotonin production and affect its CNS concentration [68]. Pseudomonas and other gut microbes synthesize serotonin from the available tryptophan and exploit the same for its virulence and intercellular signaling. In state of dysbiosis, the rate of synthesis of serotonin declines due to a reduction in the levels of circulating tryptophan, which ultimately impacts serotonergic neurotransmission. The optimal functional status of serotonin signaling is of utmost importance for neuronal networking and neurotransmission; any undesired rate of serotonin transmission impacts the functioning of the central and enteric nervous system, which has an undesired impact on gut–brain axis communication [69]. Normal microbial flora also impact brain-derived neurotrophic factor (BDNF) levels of the hippocampus, cortex, and other brain regions [70]. BDNF is an important neurotrophin and a classic trophic factor that is extensively involved in CNS growth and development by regulating structural remodeling, enhancing neurogenesis, neurotransmission, neuroplasticity, cell growth, differentiation, and overall homeostasis. AD is categorized by a decreased expression of BDNF, which may escalate Aβ and Tau (neurofibrillary tangles) pathology. The reduced BDNF expression is now recognized as a reliable marker for AD [71]. The administration of prebiotics in rats has led to the increased levels of BDNF in the frontal cortex and hippocampus, the regions governing crucial aspects of memory [59], while the decrease in BDNF and increase NFKB signaling has been seen after antibiotic treatment. NFKβ induction leads to signaling and activation of inflammatory pathways and angiogenesis have been witnessed in animal models. The administration of lactobacilli and preserving the microbiota has shown to alleviate inflammation [72]. The role of probiotics has been advocated by various studies conducted in germ-free mice which show variations in BDNF and other neurotransmitters, and thus, exhibit deficits in learning, memory, and emotional behaviors [73]. Antibiotic treatment also alters the expression of BDNF and its high-affinity receptor (TrkB) in both ENS and CNS. An interesting observation was that dysbiosis leads to regionally differentiated expression of BDNF and TrkB levels, as there was upregulation of their levels in ENS, while both BDNF and Trkb were downregulated in the hippocampus and unmodified in the prefrontal cortex [74]. Furthermore, in the latest finding, the probiotic treatment (L. plantarum IS-10506) has upregulated the BDNF expression in the hippocampus, implying that probiotics may have a beneficial role in cognition as the increased BDNF expression in the hippocampus is a verified strategy and mechanism of action of various memory enhancers, which support the fact that probiotic supplementation has positive effects on brain development and brain plasticity [75].

### 3.2. AD: Microbiota, Inflammation, and Dysbiosis

Millions of nerves endings in the GIT are crucial for organizing and coordinating intestinal functions by communicating with the brain through the vagus nerve, transmitting signals from the brain to the gut, and vice versa. The well-conserved microbiota maintains the intestinal permeability and does not allow the pro-inflammatory cytokines to travel the brain via the bloodstream [76]. Microbiota preserves intestinal permeability by regulating the epithelial barriers. In a state of dysbiosis, the altered permeability leads to systemic inflammation, which can lead to neuroinflammation by affecting the blood–brain barrier permeability [77]. Neuroinflammation and provoked IL-1, TNF-α, and TGF-β, as well as other cytokines, are involved in the neuroinflammatory response in AD. The release of cytokines by microglia and astrocytes are mediated through classical pathological markers of AD, including Aβ and NFTs. The bi-directional communication between Aβ deposition and release of proinflammatory mediators also involves genetic influence (TREM2, CD88, and D1) [55]. Gram-negative bacteria, being the major component of microbiota and lipopolysaccharides (LPS), are the major components of their cell wall. In a healthy scenario, LPS cannot reach the bloodstream due to an intact intestinal epithelial barrier, and in compromised intestinal permeability, there are increased levels of LPS in the bloodstream. Plasma levels of LPS reach 3–4 times higher in AD patients, indicating the LPS infiltration and impact of the dominance of gram-negative bacteria in AD. In the case of intestinal dysfunction and compromised blood–brain barrier, there is a huge possibility that the increased excretion of LPS may cross the intestinal barrier and reaches in the vicinity of CNS via infiltrated cytokine. The increased concentration of LPS in the fourth ventricle induces inflammation, where they are recognized by Toll-like receptors, and the accumulation of Amyloid beta is induced [78]. Microbiota also controls the maturation and development of microglia in the nervous system. A state of dysbiosis, as experienced in germ-free mice, leads to quantitative alterations in microbiota which ultimately impair immune responses and may activate AD progression [79]. Amyloidosis leading to dementia also works by the upregulation of proinflammatory cytokines (IL-6, CXCL2, NLRP3, and IL-1β). This proinflammatory scenario is accompanied by a reduction in beneficial microbiota (*E. rectale*) and an increase in harmful species (Escherichia/Shigella), leading to the conclusion of a positive relation between pro-inflammatory cytokines and the number of pro-inflammatory intestinal bacteria and a negative correlation between pro-inflammatory cytokines and the number of anti-inflammatory intestinal bacteria [80]. Neuroinflammation has an important role to play in the pathogenesis of AD. The glial cells have a physiological (ensures brain development and neuronal survival) and protective role (ensures defense against infections and harmful stimuli) in AD. This glial activation is acute and is resolved when the threat of infection is removed, but in the case of AD, this glial activation becomes chronic, and there is a constant release of proinflammatory cytokines and activation of the complement system by microglia, including IL1-β, TNF-α, and IL-6. This inflammatory storm is accompanied by a loss of microglial phagocytic activity, which results in the aggregation of neurotoxic substances such as Aβ. Aβ also fosters activation of microglia, and this vicious cycle becomes detrimental in AD, as the close association of microglial activation and cognitive deficits reveals [81].

### 3.3. Age, AD, and Microbiota

Age is one of the strongest factors for the development of AD. Age-associated changes are also intricately linked to the progression of this disorder. Dietary habits, environmental factors, concurrent infections, and drug use determine the gut composition in aged people, and alteration is accompanied by altered GI motility, malabsorption of nutrients, and impairment in immunity. All these changes in microbiota are accompanied by atrophy of the brain, impaired immune response, oxidative stress, aggregation/misfolding of proteins (Aβ), and cognitive dysfunction [58]. Age-associated hyperstimulation of the immune system leads to chronic low-grade inflammation of GIT mucosa, leading to an alteration in microbial diversity leading to increased levels of proinflammatory bacterial products in systemic circulation and their probability to cross the blood–brain barrier [82]. There are also changes in the microbial composition (levels of *Bacteroidetes*, *Lactobacillus*, and *Bifidobacteria* declines) besides the ratio of Firmicutes/Bacteroidetes, which increases to 10.9 in adulthood from 0.4 (infants), and decreases to 0.6 in elders [83]. The levels of three bacterial families, *Bacteroidaceae, Lachnospiraceae*, and Ruminococcaceae, decrease with age. Apart from this, Coprococcus, *Roseburia*, and *Faecalibacterium* decrease with age, while Oscillopsia and the Bacteroidales order (Odoribacter and Butyricimonas) increase with age. These changes are not consistent and permanent, yet it has been shown that increased age is accompanied by an alteration in microbiota and diversity [84]. These changes in the microbial population are complemented by an age-associated low-grade chronic inflammatory state and overstimulated innate and adaptive immune response that alters intestinal permeability and translocation mechanisms of bacteria. These changes of increased pro-inflammatory infiltration (IL-1, IL-6 TNF-α) are correlated with increased population of Proteobacteria and decreased short-chain fatty acid production by microbiota [85]. Thus, restoration of microbial diversity may be a therapeutic intervention in age-associated disorders.

### 3.4. Sleep Deprivation, AD, and Microbiota

Sleep disruption has both short-term (anxiety, lethargy) and chronic complications, such as cardiovascular, metabolic, and neuropsychiatric problems. Sleep deprivation is also regarded as a stressor for AD, which by modulating synaptic integrity, Amyloid, and Tau levels influences several biochemicals processes and impairs memory in AD [86]. Based on this proven evidence, altered sleep architecture is regarded as a proven etiological factor for AD and it has been suggested that disrupted sleep may promote the development of AD. Chronic sleep disruption also leads to a state of microbiota dysbiosis, and changes in the taxonomic profile accompanied by the development of insulin resistance and inflammation of adipose tissue have been noticed. The microbial diversity and diverse composition show a positive correlation with the quality of sleep, total sleep time, and inverse relation with sleep fragmentation advocating the role of healthier microbiota for healthier sleep. This microbial richness correlates well with the levels of IL-6 (a proven putative somnogen) [87]. The altered sleep architecture leads to the development of an imbalanced state of microbiota characterized by an increase in the ratio of Firmicutes to Bacteroidetes, besides a higher abundance of Coriobacteriaceae and Erysipelotrichaceae and a lower abundance of Tenericutes. A better quality sleep is also associated with a higher abundance of microbial phyla Verrucomicrobia and Lentisphaerae in stool samples. All these changes were consistently accompanied by neuro-psychological changes in AD [80]. The phyla Bacteroidetes and Firmicutes show a positive relation with sleep quality, and their effects are modulated through their influence on the circadian rhythm and food intake. Sleep deprivation alters the balanced ratio of these phyla and modulates the biochemical changes responsible for AD progression [87]. The major phyla producing γ-aminobutyric acid (GABA), a neurotransmitter that promotes sleep, are *Actinobacteria* and *Firmicutes*, while *Corynebacterium* has the metabolic ability to synthesize serotonin, a proven sleep modulator. Interestingly, various taxa from the SCFA producing Lachnospiraceae family, including *Coprococcus, Blautia*, and *Oribacterium,* shares a negative correlation with sleep [87]. The microbial diversity and composition influence sleep architecture not only through modulation of neurotransmitters and cytokine but may also impact metabolic alteration (insulin resistance), which is also an independent etiological factor for AD [88].

### 3.5. Amyloid Accumulation, AD, and Microbiota

Amyloid plaques, the crucial pathological markers of AD, are cleaved by a transmembrane protein, amyloid precursor protein (APP), which has significance for intracellular transport, neuronal signaling, and development. APP is processed by non-amyloidogenic (α and γ secretases) and amyloidogenic pathway (β and γ secretases). The amyloidogenic pathways yield Aβ peptides of different dimensions, among which Aβ40 is abundant and less toxic, while Aβ42 is abundant, neurotoxic, and forms the core of plaque-forming oligomers and protofibrils [89]. Recently, Aβ was recognized as an antimicrobial protein and part of the innate immune system. Monomeric Aβ having lesser antimicrobial property but aggregated form shows antimicrobial pore-forming structures. This process of amyloid formation involves a myeloid differentiation primary response (Myd-88) pathway, which involves activation by TLR. The Myd-88 path activates NFKB, generates TNF-α, and propagates inflammatory cellular signaling. The Myd-88 deficiency ameliorates AD amyloid aggregation in the AD animal model [90,91]. The amyloid protein synthesized in the gut helps bacterial cells to get bound to each other and forms an overlapping film that defies destruction by physical or immune challenges and ensures survival time. The bacterial amyloid and amyloid relevant to AD pathology share similarities in tertiary structure, although differ in the primary structure. Thus, bacterial amyloid in the gut may fine-tune the immune system and escalates immune response to amyloid deposits in the brain and may provoke the amyloid pathogenesis in CNS [92,93]. A huge number of microbiota strains *Mycobacterium tuberculosis, Escherichia coli, Bacillus subtilis*, etc. can synthesize amyloid protein in the gut, and this may lead to aggregation and cross linking of protein in CNS through the crossing and bypassing of the barriers [94,95]. The gut amyloid may be the source of misfolding of neuronal proteins such as alpha-synuclein and Aβ via cross seeding that ultimately prime the innate immune system and signal neuroinflammation [96]. The higher secretion of Aβ by bacteria and fungi disturb the dynamic equilibrium and may escalate the risk of AD [24]. This may be an explanation of concurrent infections in the case of AD and the effects of lifestyle modifications and food habits on these pathologies.

### 3.6. Microbiota, Inflammation, and Oxidative Stress

Oxidative stress, mitochondrial deficits, and bioenergetic failure is a well-characterized pathological feature of AD [97]. Microbiota also has an important role to play in these functions. The intestinal tissues (neural plexus, smooth muscle, and mast cells), commensal anaerobes, and leukocytes are rich sources of Nitric oxide (NO). High nitrate intake increases nitrite as well as NO production and this function is aided by *Lactobacilli* and *bifidobacteria* (Convert nitrate and nitrite in NO) and gut *streptomycetes* and *bacilli* (produce NO through their NO synthetase from L-arginine) thereby increase the level of NO by host epithelial cells [98]. NO is the major neurotransmitter of noradrenergic and noncholinergic ENS and is also released by glutamate-mediated activation of NMDA receptors. In minor concentration/optimum levels NO is neuroprotective, while higher levels induce oxidative stress-induced damage, becomes a neurotoxin, and activate apoptotic pathways at higher levels. NO also induces mitochondrial dysfunction. SFA produced by microbiota reduces mitochondrial defects, ROS levels, and ROS-induced DNA damage and activates the antioxidant mechanism ensuring cell survival. In a state of dysbiosis, when levels of pathogenic Bactria such as Salmonella and E. coli are increased, there is excessive production of H2S, which alters the metabolism, increases lactate levels, reduces ATP production, and signals the inflammatory pathways (IL-6). The abundance of Proteobacteria and decrease of Bifidobacteria in case of dysbiosis leads to decreased production of SCFA developing into dyslipidemia. Bifidobacteria is the major taxon that has a lipid-lowering and hypocholesterolemic effect by facilitating fecal elimination of cholesterol and reducing its absorption. This taxon also leads to an increase in serum levels of leptin, an anti-obesity hormone. Leptin has shown an association with LTP and pro-cognitive effects; thus, it may have added advantage in normalizing AD pathology. This has been further supported by the report of a probiotic formulation (Lactobacilli and Bifidobacteria), which not only normalized the serum triglyceride levels and improved cognitive performances, but also confirms and supports the role of microbiota in the maintenance of a balanced lipid metabolism. Lacctobaccili also reduces ammonia levels. Ammonia, which is neurotoxic at minor concentrations, produces mitochondrial deficits and is implicated in AD. Lactobacillia maintains optimum Glutamine synthetase, SOD, and Glutathione peroxides activity, which reduces ammonia levels and helps to escape the detrimental age-associated effects of ammonia [13].

### 3.7. Dietary Intervention in AD and Microbiota

The benefits of omega 3 polyunsaturated fatty acids (docosahexaenoic acid, eicosapentaenoic acid) in AD are credited to their antioxidant potential, but it has been revealed that intestinal absorption of these food subtypes is affected by intestinal surroundings and contents of fatty acids in the diet. Thus, the absorption status and capacity of the intestine decide the content and potency of these fatty acids. Additionally, consumption of coffee, which is proposed to have beneficial results in AD, is indebted to its mechanism at various levels (phosphodiesterase inhibition, CREB modulation, antioxidant potential), but it also works at the level of the microbiota. The fibers in coffee are digested by gut bacteria, who harvest energy for their own growth. It also reduces the ratio of Firmicutes to Bacteroidetes, which is associated with reduced inflammation [99]. The polyphenolic content of coffee beans is degraded by microbiota into easily absorbable small molecules, and thus, complete utilization of these polyphenols for their biological activity is possible only through an intact bacterial population [100]. Further higher intake of saturated fat induces Aβ accumulation and microgliosis and is associated with AD risk, while calorie restriction has the opposite effects. These changes are attributed to the capability of a fatty acid-induced diet to alter the microbe, change the intestinal permeability, and increase LPS absorption. These inflammatory changes induce endotoxemia-mediated systemic inflammation, while restricted calorie intake optimizes gut microbiota composition and diversity [58]. Microbiota influence the gut epithelia and immune system. The inflammatory contribution of microbiota is attributed to metabolic endotoxemia and its influence over the modulation of neurohormones, triglyceride clearance, and mucosal immunity. Microbiota influence bioenergetics through their capacity to harvest energy from food and their impact on modulation of lipopolysaccharides levels, which have a role in inflammation and can contribute to several metabolic (type 2 diabetes, obesity) and neurodegenerative complications [101].

### 3.8. AD, Blood Brain Barrier, and Microbiota

Gut microbiota also influences the integrity and development of the blood–brain barrier. GF mice show increased permeability of BBB compared to specific pathogen-free mice and mice with normal gut microbiota. Reconstitution and normalization of the gut microbiota decreased BBB permeability and upregulates the expression of tight junction proteins [102]. Moreover, the integrity of BBB is altered by the overactive immune system as results of dysbiosis. The microbial metabolites and cellular components which emerge from a state of dysbiosis first modulate the innate and adaptive immunity, but in later stages, may signal neuroinflammation through the activation of mast cells, ROS, cytokines, and chemokines. These cytokine components influence BBB increases CNS infiltration through the involvement of astrocytes, microglia, and the vascular system of the CNS. The peripheral respiratory infection by *Bordetella pertussis* is known to promote T cell infiltration, neuroinflammation, and aggregation of Aβ. In a 5xFAD model, a state of dysbiosis led to changes in Aβ-related pathogenesis and promoted gliosis. This over-activation of microglia was suppressed by antibiotic treatment. Further degradation in BBB increases the chances of peripheral infection to reach the CNS and influences AD pathogenesis and Aβ formation. These findings are further supported by research that has demonstrated that CNS-invading fungi Candida albicans and viral infection in the brain by Herpesviridae dramatically accelerated Aβ deposition with concurrent gliosis [103]. These findings get enough support from novel therapies targeting AD that include antiviral drugs (Acyclovir, Penciclovir, Foscarnet, Bioflavonoids, and Valacyclovir) and antimicrobial drugs (Doxycycline, Propranolol, Rifampicin, and Gingipin inhibitors) which have shown neuroprotection in animal models of AD [104].

## 4. Stress in AD

A plethora of evidence supports a complex interaction between CNS/ENS and microbiota in the pathological domain. These include comorbidity of GIT disorders and psychiatric illnesses, mood disorders, and irritable bowel syndrome. The effects of genetic and environmental factors are modulated by the gut–brain axis, which ultimately affects brain development and function; thus, they are implicated in a variety of CNS complications. Interestingly stress is an important factor that can trigger depression and anxiety and influence pathological progression of IBD, and in all these pathologies, microbiota modulation is involved. Thus, it seems appropriate to add that microbiota may have a role in these disorders [44]. Among various factors, lifestyle factors, including exposure to high-pitched noise, insomnia, altered pattern in circadian rhythm, and sedentary behavior, are found to affect the pathogenesis of AD. Interestingly, all these factors are also found to alter the microbiome diversity and status. Thus, it may be predicted that the external stressful events may trigger or modulate AD pathogenesis through the microbial route, as microbial density is highly susceptible to these obnoxious and undesirable changes [80]. Thus, elucidation of the linkage between modern lifestyle/stressful events, gut microbiome, and AD is a feasible area that requires special mention. Stress is a crucial modifiable factor that intermingles with the physiological mechanisms that may activate the AD cascade. Stress is a pervasive feature that encompasses environmental, physical, and psychological components. The biological effects of stress involve and include the core stress response system, the hypothalamic-pituitary-adrenal (HPA) axis, the overactivation of which leads to abnormal and maladaptive changes in brain anatomy and physiology [105]. Stress is a conceivable threat to homeostasis which provokes a physiological, psychological, and adaptive response in an individual. Stress is resolved when the incoming or inducing threat is resolved, but chronic stress is a determinant of human health [97]. The commons stressor in the chronically activated state has found an association with imbalanced immune activity (autoimmunity or overactivation), endocrine disruption, inflammation, and psychological and cognitive impairments. In an acute or transient state, all these changes may help to restore homeostasis, but in the aggravated state, they may turn hostile and detrimental [34]. The physical and psychological stress mechanisms involve the HPA axis, the sympathetic nervous system, and the inflammasome-mediated induction of inflammatory responses. Although sources of stressors may divulge, many of the biochemical mechanisms triggered as part of the stress response are common for physical and psychological stressors. AD is a disorder of varied origin and diverse etiopathology. Stress is an inescapable event that, when within limits, helps to restore homeostasis, while in uncontrolled form, it can disturb the hypothalamic–pituitary–adrenal (HPA) axis. The HPA axis is integrally involved in cognitive development; thus, when in an aggravated form, it leads to neuroinflammation, dysfunctional immune system, hippocampal atrophy, and intellectual and neuropsychiatric impairments [97]. A stress-activated HPA axis results in the elevation of Glucocorticoids (GC) levels (cortisol in humans and corticosterone in rodents). The corticotrophin-releasing hormone stimulates ACTH, which in turn increases the production of glucocorticoids from the adrenal cortex. The glucocorticoid hormone exerts a negative feedback on the HPA axis. In stressful situations, cortisol secretion continues for hours, until a critical concentration in the blood is reached and cortisol inhibits CRF and ACTH release by the hypothalamus and the suprarenal gland, respectively, establishing a negative feedback loop that allows the recovery of internal homeostasis [106]. The mineralocorticoid receptors (MRs) and glucocorticoids receptors (GRs) are the two major receptors to bind with Glucocorticoids. Interestingly, both MRs and GRs are highly expressed in pyramidal neurons of CA1 and CA2 and in granule cells of the dentate gyrus of the hippocampus, and long-lasting stress (hypercortisolemia) could be a potential neurodegenerative factor for the hippocampus. The continuation of stressful events and hypercortisolemia activates inflammatory responses and the corresponding dysregulation of inflammatory feedback [107] and reduction of synaptic plasticity and neurogenesis, a critical processes for the prevention of allostatic load [108]. GCs also influence the accumulation of Aβ, the formation of NFTs, dendrite atrophy, and downregulation of matrix metalloproteinase-2 (MMP-2) (involved in clearance of plaques) [105] (Figure 3).

### Stress and Dysbiosis

A diverse gut microbiome is healthier and is also associated with better learning/memory abilities, optimum behavior, and healthier cognitive functions [109]. As already discussed, the microbiota has a unique and individualistic profile and the functional, physiologic, and flexible status of an individual is altered by several factors, including stress. Stress influences microbiota composition and function and microbiota composition influences stress responses, making this reciprocal relationship an extremely complicated yet viable one to investigate stress and microbiota relationship in AD [110,111] (Table 1). The signals of microbiota are transmitted to brain/peripheral tissues that secrete glucocorticoids. Although these steroids are secreted from the adrenal cortex, they can also be secreted from the intestine via regeneration of biologically active glucocorticoids, corticosterone, or cortisol from their inactive 11-oxo derivatives by enzyme 11b-hydroxysteroid dehydrogenase type 1 (11HSD1), constituting extra-adrenal glucocorticoid synthesis. The glucocorticoid secretion also occurs through neuropeptides, cytokines, and even bacterial ligands independently of pituitary ACTH. Therefore, it is conceivable that gut microbiota might affect the steroidogenesis of glucocorticoids [112]. Additionally, the microbiota is influenced at various stages by the hypothalamic adrenal axis (stress axis). At the time of activation of the HPA axis, microbiota starts to take a final shape, as it is innervated by the HPA axis through the enteric nervous system under the influence of this axis [32]. The altered microbiota influences the HPA axis, which modulates the stress response, which has been exemplified in various studies related to the development of anxiety and social behavior in rodents in a state of dysbiosis. There was upregulation of c-FOS levels (responsible for anxious state) and alteration in brain-derived Neurotrophic factor (BDNF) in the infectious state induced by stress, and all these changes were normalized by probiotics treatment [60]. Stress modulates the microbiota in several ways, which include redirecting blood away from the GIT (decreasing oxygenation), modulating immune functions of the GIT, decreasing the availability of substrates, decreasing GI motility, decreasing the production of antimicrobial peptides, and producing an oxidative stress and inflammation-prone environment [30,34]. Various studies advocate microbiota as stress mediators, and studies in germ-free mice have shown a many-fold increase in the stress response (increased corticosterone/ACTH secretion), suggesting a calming/soothing effect of microbes on the devastating effects of the HPA axis, which include neuroinflammation, cognitive deficits, and altered behavior [6]. A chronic stressor is regarded as a promoter of dysbiosis and normalizing the flora using probiotics and prebiotics has been reported to reduce stress-induced corticosterone levels and has shown beneficial effects on anxiety and depression through the modulation of cytokine levels and production of short-chain fatty acids by microbiota [113]. Supplementation of lactobacillus has also shown a decrease in corticosterone levels and benefits in cognition. The increased concentration of the Clostridiales family and decrease in the abundance of Bacteroides has been observed in different chronic stress paradigms, which also correlated with the levels of proinflammatory cytokines [114]. Collectively, these data support the influence of stressful events on microbiota density. Chronic antibiotic use also leads to dysbiosis with a consequent increase in serum corticosteroid levels. These changes were accompanied by increase inflammation and cognitive abnormalities, which were reversed by the recomposition of microbiota through probiotics [55]. Antibiotic use of ampicillin in rats for one month resulted in dysbiosis, elevated corticosterone levels, impairment of behavioral spatial memory, and development of anxiety. These changes were reversed by Lactobacillus fermentum [55]. In another interesting study, the gram-negative *Citrobacter rodentium* produced changes in microbial function, which produced cognitive deficits only in concurrent presence of psychological stress; probiotic prophylactic use restored the gut balance and reversed the stress-induced cognitive changes [115]. Intestinal secretion of hormones and neurotransmitters (catecholamines) due to increased stress modulates microbial growth. Additionally, the altered signaling of ENS and vagus nerve modulates physical forces which reduce digestive activities and GI motility, and as a result, substate availability. A cycle of ischemia, hypoperfusion, and reduced oxygenation is also experienced due to diversion of blood, which reduces oxygenation of the intestinal mucosa and may lead to hypoxia-related inflammatory cascade, immune overactivity, altered membrane permeability, and ultimately a change in metabolism and microbiota [34]. Cortisol secretion in response to stress alters the immune activity and microorganisms also sense the stressors and deviate towards a change in their composition. The detection of changes in stress levels/hormones helps microbes to change their gene profile to suit the need of individuals and by this mechanism; they also control their own population. Recently, it has been proposed that Ruminococcus, a member of the gut microbiome in piglets, controls levels of n-acetyl aspartate, the second most abundant molecule in the brain, via levels of cortisol [116]. The germ-free mice exposed to acute restraint stress and acute novel environment stress show exaggerated stress response which was normalized with colonization with commensals and treatment with prebiotics and probiotics [111]. Germ-free mice with concurrent upregulation of stress response have shown a reduction in BDNF levels, alteration in NMDA levels, and have increased anxiety with the deterioration of neurological functions [53,65]. The maternal separation stress in neonatal rats also leads to long-term behavioral changes with concurrent changes in gut microbiota [117]. Normally, the small quality of antigens crosses the epithelial barrier and interacts with the immune system for human protection, but stress compromises epithelial functions and mucosal defense, which ends with signaled infiltration of antigens which overactivated the immune system and systemic inflammation [118]. Additionally, the physiologically sterile epithelium is altered in exposure to stress. There is a significant decrease in fecal bacteria, especially Lactobacilli, and this alteration produces the same changes as the inhibition of gastric acid release, alterations in gastrointestinal motility, or increased duodenal bicarbonate production. Overall stress may lead to conditions less favorable for the growth and proliferation of beneficial bacteria such as Lactobacilli, as well as reducing survival and adherence [119]. The stress-mediated multi-fold increase in catecholamines affects intestinal functions and microbiota in many ways. On the contrary, probiotics such as Lactobacillus reduce plasma corticosterone levels, subside inflammation, restore serotonin levels, and increase BDNF levels, culminating in precognitive effects [120]. Stress favors the colonization of pathogenic bacteria via the alteration in secretions of intestinal secretory IgA, leading to a dysfunctional gut–brain axis [78]. A recent study shows that chronic elevation of GC changes the biodiversity of microbiota by increasing the colonization of pro-inflammatory taxa including Proteobacteria. Stress exposure also decreases *Clostridiales* and *Lactobacillus*, related to inflammation. The RT-qPCR analysis of 16S rRNA gene abundance in fecal microbiota indicates that GC exposure reduces the levels of anti-inflammatory bacteria. Although GC itself has a strong anti-inflammatory effect, long-term exposure produces an adverse effect, by acting on the gut microbiota [121]. Stress or GC exposure also alters the circadian clock and peripheral oscillations. Long-term exogenous dexamethasone administration resulted in rhythm loss, which may lead to the development of metabolic syndrome-impaired lipid metabolism and inflammation [122]. Further mucus production is also reduced by Chronic GC levels, as shown by a reduction in the expression levels of Muc2 and Muc3. The mucus layer ensures the stability of the gut microbiota and changes in mucus secretion alter bacterial gut homeostasis. Experimentally, dexamethasone treatment upregulated the expression levels of colonic antimicrobial peptide-related genes, such as Tlr-2 and Defa8, which is also a direct factor that can alter bacterial structure [121,123]. Through this discussion, it seems appropriate to conclude that stress is an interesting player to induce dysbiosis and to proceed to related pathological progression of AD (Figure 3).

## 5. Targets

### 5.1. Ferulic Acid

Ferulic acid (FA) is a phenolic compound densely found in seeds and fruits of plants and in vegetables. FA is also synthesized by gut microbial species, including L. fermentum NCIMB 5221. FA is an excellent free radical scavenger supplemented by anti-inflammatory action. FA has also shown anti-aging, neuroprotective, and neuroregenerative properties. It has also shown anti-amyloid properties and the ability to upregulate BDNF and NGF levels that may be of paramount importance in AD [130,131,132]. The majority of dietary polyphenols is not absorbed by the small intestine and gets aggregated in the colon where microbial species metabolize them to absorbable phenolic acids. The phenolic compounds have antioxidant and anti-inflammatory properties, and several of them, including caffeic acid and ferulic acid, are bioactive in inhibiting the generation of beta-amyloid (Aβ) peptides. Their anti-amyloid and anti-inflammatory role proven in experimental studies show that the modulation of microbial pathways may be targeted to unleash the bioactive potential of these phenolic compounds [133].

### 5.2. Histamine

Histamine is one of the important neurotransmitters that regulate neuromodulatory mechanisms important for behavioral alterations. Histamine also has an important role in immunomodulation, allergic reactions, and inflammatory cascade. Many species, such as Pediococcus, Lactobacillus, Lactococcus, Streptococcus, and Enterococcus, can secrete histamine and regulate biologically important levels of histamine [134]. Several probiotics, such as L. reuteri, exert their immunomodulatory activity and suppress TNFα levels via histamine production, which reduces activation of TLR-mediated pathways important for cytokine modulation [135]. Gut histamine is known to have a potential therapeutic effect in neurodegenerative diseases such as AD by inducing allergy or anti-inflammatory responses by acting on H4R receptors. However, the CNS responses are receptor-specific [132].

### 5.3. Ghrelin

Ghrelin is an important biomolecule that affects neurological homeostasis and neurological functions. The levels of Ghrelin can be modulated through microbiota. It acts as a satiety hormone and a neuropeptide in the CNS. It is produced when the stomach is empty to ease the hunger sensation. Ghrelin has an important role in energy homeostasis, neuroprotection, and immunomodulation. Gut microbial dynamics affect ghrelin production, and its levels are decreased in the state of dysbiosis (i.e., alteration in populations such as Bifidobacterium spp.) [136]. Ghrelin shows beneficial effects in experimental models of neurodegenerative complications owing to its antiapoptotic, antioxidant, and anti-inflammatory properties, which are complemented by its ability to improve neuronal plasticity and antiaggregatory role on amyloid-β. In an experimental setup, exposure to *Lactobacillus acidophilus*, *Lactobacillus casei*, *Bifidobacterium bifidum*, and *Lactobacillus fermentum* for 12 weeks showed the precognitive effects through the modulation of ghrelin levels [45].

### 5.4. BDNF

GF mice also had lower brain-derived neurotrophic factor (BDNF) expression, a protein which is important for neuronal growth and synaptic plasticity [59]. Probiotic supplementation has shown the promotion of neurogenesis and normalization of the HPA axis. The increased levels of BDNF improve nutritional status, show the antioxidant property, and decrease the expression of inflammatory cytokines [137]. Mice deficient in BDNF show altered development of the GI tract innervations and reduced expression of BDNF was found to be specifically associated with increased anxiety and progressive cognitive dysfunction [37]. The probiotic supplementation of VSL#3 (a combination of eight different strains, namely *Streptococcus thermophilus*, *Bifidobacterium breve*, *Bifidobacterium longum*, *Bifidobacterium infantis*, *Lactobacillus acidophilus*, *Lactobacillus Plantarum*, *Lactobacillus paracasei*, and *Lactobacillus delbrueckii* subspecies Bulgaricus) to aged rats induced a robust perturbation in gut microbiota composition, which was accompanied by gene expression changes in the brain cortex, attenuated age-related deficits in long-term potentiation, decreased microglial activation, and increased BDNF and synapsin levels [138]. The deficiency of BDNF in germ-free mice not only produces cognitive deficits resembling pathogenesis of AD, but there is also impairment of vagal sensory innervations, indicating the involvement of gut microbiota in the regulation of BDNF expression [58].

### 5.5. Silibinin and Silymarin

Silymarin isolated from milk thistle has been implicated beneficially in liver ailments. A recent study has found a correlation between silymarin and its active constituent silibinin in AD pathology. Although accompanied by drawbacks of poor bioavailability, further studies are needed to affirm the claim. In a recent study, silibinin and silymarin alleviated cognitive deficits and reduced plaque burden in the brain of APP/PS1mice. Silymarin and silibinin altered the microbiota composition in such a way that there was upregulation of the Akkermansia and Allobaculum genus in microbiota, and the beneficial effects may be attributed to the decrease in D-alanine levels in the AD brain. Silymarin and its main active component silybin administration could alleviate the memory deficits and reduce amyloid plaques in APP/PS1 mice [139]. Silymarin has also shown beneficial results against oxidative stress-induced neuronal injury, ethanol-induced neurotoxicity, reduced Aβ levels, and microglial activation and showed behavioral benefits in AD. Most benefits may be attributed to its upregulation of BDNF levels, antiapoptotic, neurotrophic, antioxidant, and anti-inflammatory properties, and regulatory check on neurotransmitter functions [140].

### 5.6. SCFA

The decreased diversity of microbiota is well documented in AD patients (greater abundance of pro-inflammatory bacterial taxa along with fewer anti-inflammatory taxa [54]). Among various beneficial metabolites produced by microbiota, short-chain fatty acids (SCFA) should be given extra attention due to their diverse roles. SCFA are the key mediators along the gut–brain axis, resulting in increased microglial activation and ApoE upregulation, and Aβ deposition of SCFA includes acetate (which regulates food intake and levels of GABA and glutamine), propionate, and butyrate. Butyrate is a major chemical that provides energy to intestinal cells, inhibits histone deacetylases, and alters the intestinal and neuronal transcription of genes. The dietary-rich supplementation of butyrate amplifies memory-related genetic programs and restores DNA acetylation and has shown benefits in AD pathogenies [141]. All three metabolites are present in human CSF and the concentration of butyrate is increased in the human brain by live Clostridium butyricum. SCFA has been reported to cross the blood–brain barrier and has a role in neuronal homeostasis. Germ-free animals have shown reduced expression of tight junction proteins and have defected BBB permeability. Propionate and butyrate also have a role in the regulation of intracellular potassium level, cell signaling, and in the regulation of levels of Serotonin, Dopamine, and Nor-adrenaline; thus, they influence the beneficial effects of SCFA, which may be attributed to the upregulation of BDNF, NMDA levels, and their effects on NPY levels and serotonin transporters. Studies on chronic psychosocial stress have also shown a possible application for prebiotics and SCFAs in reverting cognitive and neuropsychiatric effects by reducing stress-induced corticosterone release [142].

### 5.7. Ketogenic Diet (KD)

The KD refers to a diet that may produce physiological ketosis/ketonemia (increase in ketone bodies in blood). The KD (a diet rich in fat and proteins and poor in carbohydrate content) is well recognized for its antiepileptic role in pediatric epilepsy has also shown beneficial results in AD. The KD shows anti-inflammatory effects, decreases apolipoprotein E glycation, modulates neurotransmitter levels, improves mitochondrial deficits, increases insulin sensitivity, and alters the state of hypometabolism (GLUT-1 deficiency) [143,144] prominent in AD [95]. Many of these effects are attributed to neurotransmitter modulation, epigenetic modifications, and altered microbiota [145]. A ketogenic diet has been reported to increase the abundance of intestinal beneficial taxa (*Akkermansia muciniphila* and *Lactobacillus*) and reduce the abundance of intestinal pro-inflammatory taxa (*Desulfovibrio* and *Turicibacter*) [146].

Ketones bodies, acetoacetate, and β-hydroxybutyrate, which are the main ketone bodies, have been proven to be neuroprotective and could contribute to the overcoming of the cognitive deficits that occur with aging through an increase in capillary density and levels of hypoxia-inducible factors. Considering diet as the main modulator of gut microbiota, keto microbiota could be a key factor involved in keto-therapeutic effects, since the microbiome is a vital network between different organs [147,148]. In AD, the compromised glucose metabolism correlates with the cognitive performance and the hypometabolic state is prominent years before the clinical presentations. Elevated ketone levels improve total energy metabolism in humans, which has also shown positive effects on cognitive performance [149].

## 6. Future Perspectives and Conclusions

Several preclinical and clinical studies have documented the benefits of microbial modulation in several gastrointestinal, endocrinal, metabolic, and neurodegenerative complications. Microbiota also influence the neuropathological cascade of AD and microbiota modulation has shown encouraging results, including the reduction of amyloid plaques and neurofibrillary tangles, alleviation of glial responses, and improvement in cognitive performance. Additionally, in transgenic animals, fecal microbiota transplantation has been shown to improve synaptic plasticity, decrease plaques and inflammatory mediators, cause an overall improvement in pathological markers, and improve cognition [150,151]. The bidirectional communication between gut and brain, the concurrent GIT and neurological disorders, and earliest evidence of beneficial effects of laxatives and oral antibiotics in hepatic encephalopathy has opened new areas of opportunities in the treatment of neurodegenerative disorders. Although it is still premature to decide whether neurodegenerative and neuropsychiatric alterations precede gut dysfunction and dysbiosis or whether gut dysfunction and dysbiosis precede brain and behavioral changes, altered microbiota in these disorders makes a strong point that the modulation of microbial identities may be an option to treat these complicated pathologies. In AD, altered microbiota-associated inflammation and downregulation of trophic factors and pro-survival signaling pathways make it a domain worth investigating. Microbiota modulation through probiotics has shown beneficial effects not only on its pathological markers (Aβ, NFTs, oxidative stress, gliosis), but activation of pro-survival mechanisms (BDNF), and their effects on neuronal chemistry makes the concept fascinating. The exaggerated stress response in mice in a dysbiosis state and in germ-free mice may help us to conclude that microbiota diversity and composition may have a role in the calming effect of reconciliation of stress effects. Stress, which is an inevitable, yet modifiable factor, affects the pathological progression of AD through diverse mechanisms, including producing a state of dysbiosis. The desirable changes produced by microbiota and pro/prebiotics on HPA/Stress axis and as a result on AD progression provides evidence enough to explore it further. The microbial influence on immunomodulation, oxido-reductase status, inflammation, and neuroendocrinal mechanisms speaks volumes about the importune of healthy microbiota for physiological homeostasis and as a therapeutic target in neurodegeneration. The interaction and interlinks of microbiota to the stress axis expand the domain of the gut–brain axis to the gut–brain–stress axis, and these results are complemented by the proven role of probiotics in the modulation of these pathways and benefits in AD pathology. In the future, meta transcriptomic and metabolomic techniques may be an asset to explore the effects of the probiotic intervention on gut microbiota in the host. Although probiotics have been widely promoted among the public, there have been some contradictory results, and the adverse profiles (gene transfer from probiotics to normal microbiota, harmful metabolic effects of probiotics, and immune system stimulation) need attention.

## Figures and Tables

**Figure 1 biomolecules-11-00678-f001:**
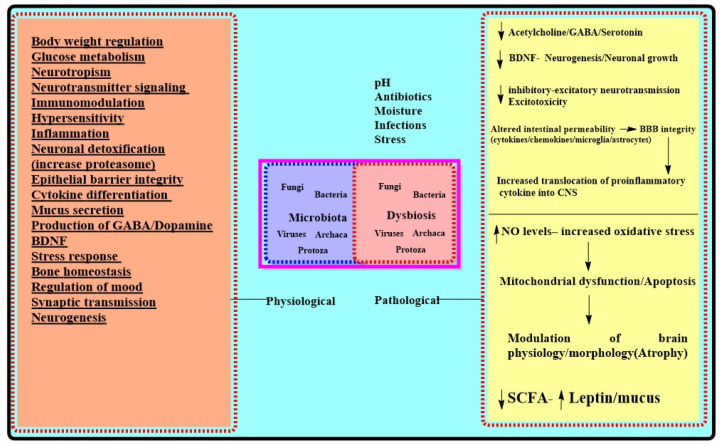
Physiological and pathological role of microbiota. Microbiota influences several physiological functions, while an altered state (dysbiosis) due to stress and other changes leads to several neurological alterations, which proves detrimental for CNS homeostasis. GABA; gamma Amino Butyric acid; BDNF: Brain derived neurotrophic factor; BBB; blood–brain barrier; CNS: central nervous system; SCFA; Short chain fatty acids.

**Figure 2 biomolecules-11-00678-f002:**
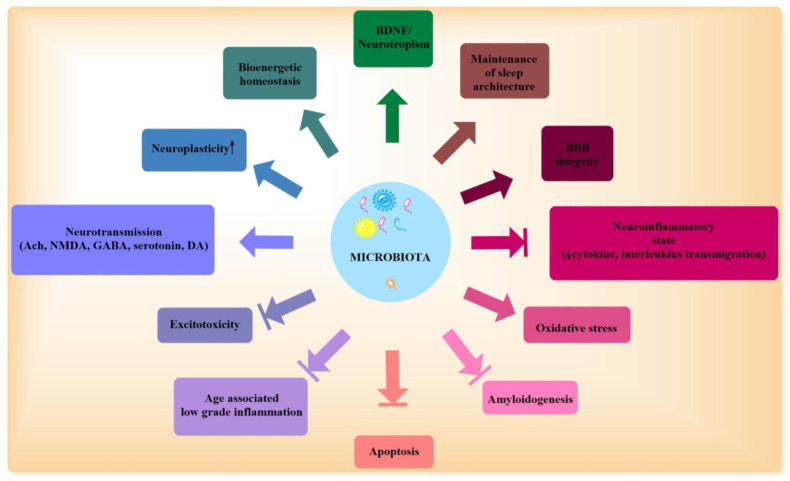
Physiological targets of microbiota modulation. Microbiota may be modulated to achieve various beneficial outcomes relevant to Alzheimer’s pathology.

**Figure 3 biomolecules-11-00678-f003:**
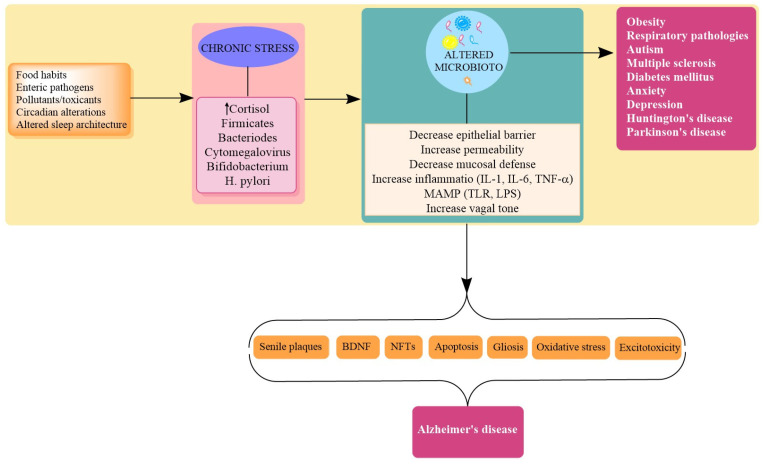
Chronic stress-induced dysbiosis leads to Alzheimer’s pathology. Chronic stressors, such as altered food pattern, sleep architecture, and bacterial infections, change the microbiome and alter microbiota. The state of dysbiosis may progress to pathological changes (NFTs, Aβ), leading to the development of Alzheimer’s disease. H. Pylori; *Helicobacter pylori*; Il-Interleukins; TLR; Toll like receptor; LPS; Lipopolysaccharides; MAMP: Microbe-Associated Molecular Pattern.

**Table 1 biomolecules-11-00678-t001:** Summary of the physiological and stress-mediated pathological outcome of dysbiosis for Alzheimer’s disease.

Physiological Role for Microbiota [23,34,38,51,75]	Effects of Chronic Stressors on Microbiota Mediated Functions [30,34,109,110,111,124,125,126]	AD Centric Pathologic Outcomes of Dysbiosis/Chronic Stress
Nutrient uptake GIT development Regulation of appetite Regulation of Mood Regulation of circadian rhythm/sleep architecture Neuronal plasticity Intestinal permeability Modulation of stress response Regulation of HPA activity Regulation of trophic factors Immune signaling Neurogenesis Glucose metabolism Production of, GABA and monoamines Synaptogenesis Synaptic signaling Excitatory/inhibitory neurotransmission	Increase in harmful bacteria (*Escherichia*, *Shigella*, *Proteus*, *Klebsiella*) [12,13] Decrease in symbionts (*Lactobaccilus*, *Proteobacteria*, *Actinobacteria*, *Bifidobacterium*, *Cyanobacteria*) [15] Increased Glucocorticoids levels Dysfunctional HPA axis [22,106] Cytokine permeation [49,50] Disrupted neurotransmitters [59] Increased cFOS/p- Glycoproteins [60,126,127] Increased MAMP (LPS) [78] Decreased BDNF/trophic factors [60,70] Increased NFKB signaling [62,128,129] Decreased SCFA (butyrate, propionate, and acetate) [40] Disrupted circadian rhythm (altered melatonin levels) [87]	Aβ aggregation [53,92] Tau hyperphosphorylation [103] Altered BBB permeability [100] Decreased neurogenesis [122] Morphological alterations in brain [103] Neuroinflammation/Microglial activation [79,123,124,125] Decreased Serotonin/5-HT/DA functions [65,66] Decreased hippocampal volume [71,124] Insulin resistance [13] Apoptotic cell demise [127,129] Oxidative stress [13]

GIT: Gastrointestinal Tract; HPA: Hypothalamic pituitary adrenal axis; GABA: gamma-Aminobutyric acid; BDNF: Brain-derived neurotrophic factor; MAMP: Microbe-Associated Molecular Pattern; NFKB: Nuclear factor-κB; SCFA: *Short-chain fatty acids*; 5-HT: 5-hydroxytryptamine; DA: Dopamine.

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
