# Peer review of "Dysbiosis and Alzheimer’s Disease: A Role for Chronic Stress?"

_biomolecules, 2021, doi:10.3390/biom11050678_

Round 1

Reviewer 1 Report

The review by Ana Margarida Pereira et al, provides an overview about the various mechanisms through which stress modulates the microbiota and advocates stress-mediated dysbiosi to target Alzheimer’s pathology. The review is clearly written, its original and of interest in its field.    I recommend that the review be accepted with minor revision:   a) The authors should better emphasize the conclusions. Also future perspectives can be provided to stimulate the clinical interest.   b) The authors should better rewrite the aim in abstract section   c) Please expand the introduction section or add a little paragraph with more studies about the relationship between gut and brain (not only AD). Please referee doi: 10.1096/fj.201901584RR and 10.3390/ijms21218136.   d) The authors should correct "microbioto" to "microbiota" in Figure 2.

Author Response

The review by Ana Margarida Pereira et al, provides an overview about the various mechanisms through which stress modulates the microbiota and advocates stress-mediated dysbiosi to target Alzheimer’s pathology. The review is clearly written, its original and of interest in its field.    I recommend that the review be accepted with minor revision:  

Query a) The authors should better emphasize the conclusions. Also future perspectives can be provided to stimulate the clinical interest.  

Response: The conclusion and future perspective already provided have been revamped and rewritten. The changes have been incorporated in the manuscript and highlighted in red

  1. b) The authors should better rewrite the aim in abstract section  

Response: In line with comment we have edited the manuscript.

  1. c) Please expand the introduction section or add a little paragraph with more studies about the relationship between gut and brain (not only AD). Please referee doi: 10.1096/fj.201901584RR and 10.3390/ijms21218136.  

Response: The section has been expanded and highlighted in the introduction section (in Red) and, the desired references have been added in the manuscript (reference No.22 and 23).

  1. d) The authors should correct "microbioto" to "microbiota" in Figure 2.

Response: In line with above comment we have edited the manuscript

Reviewer 2 Report

The submitted review is very well written and plausible with the requirements of the Journal. The subject is relevant and timely, authors have broadly discussed and presented state of the art in this emerging new discipline on gut-brain axis.

However, I suggest editing the part of the text related to the role of ketogenic dietary pattern in Alzheimer’s disease (AD), as well as in other neurological disorders and brain injuries. Namely, the role of keto bodies such as acetoacetate and β-hydroxybutyrate are products of the normal metabolism of fatty acid oxidation, which serve as metabolic fuels in extrahepatic tissues  needs to be mention in this paragraph. The description should include the unique features of these keto bodies in the energy metabolism of nerve cells. The authors are also encouraged to include the following citation in the manuscript (https://doi.org/10.2478/acph-2019-0051) when discussing the influence of the diet and biologically active compounds from food. 

Additionally, minor points:

a) Descriptions (captions) associated to the figures should be revised and more comprehensively ones provided;

b) Table should be put in the "landscape" layout.

After these revisions the manuscript can be published in the Journal.

Author Response

The submitted review is very well written and plausible with the requirements of the Journal. The subject is relevant and timely, authors have broadly discussed and presented state of the art in this emerging new discipline on gut-brain axis.

However, I suggest editing the part of the text related to the role of ketogenic dietary pattern in Alzheimer’s disease (AD), as well as in other neurological disorders and brain injuries. Namely, the role of keto bodies such as acetoacetate and β-hydroxybutyrate are products of the normal metabolism of fatty acid oxidation, which serve as metabolic fuels in extra hepatic tissues needs to be mention in this paragraph. The description should include the unique features of these keto bodies in the energy metabolism of nerve cells. The authors are also encouraged to include the following citation in the manuscript (https://doi.org/10.2478/acph-2019-0051) when discussing the influence of the diet and biologically active compounds from food. 

Response: The desired section has been added with the details and changes has been highlighted (in green). The desired article has been included (reference No.101) and changes highlighted in Green.

Additionally, minor points:

  1. a) Descriptions (captions) associated to the figures should be revised and more comprehensively ones provided;

Response: In line with above comment we have updated the captions

  1. b) Table should be put in the "landscape" layout.

Response: we have added the table in landscape layout in revised manuscript.

After these revisions the manuscript can be published in the Journal.

Reviewer 3 Report

The review paper has been submitted to the journal of biomolecules-1170267. The paper is very interesting and contains tremendous information related to dysbiosis and Alzheimer disease.

The paper is well-written, and the authors attempted to cover the topic thoroughly. The review is straightforward to follow and giving an impression to explain and clarified various mechanisms related to stress-mediated dysbiosis.

My impression is that the topic is important getting attention since the dysbiosis is or can be associated to several essential functions in AD or neurodegenerative disorders.

I enjoyed reading the review and learnt many issues related to Alzheimer’s disease and dysbiosis. The paper gave an impression for further studies and missing links in the AD pathogenesis, which must be reconsidered by many researchers.

Author Response

The review paper has been submitted to the journal of biomolecules-1170267. The paper is very interesting and contains tremendous information related to dysbiosis and Alzheimer disease. The paper is well-written, and the authors attempted to cover the topic thoroughly. The review is straightforward to follow and giving an impression to explain and clarified various mechanisms related to stress-mediated dysbiosis. My impression is that the topic is important getting attention since the dysbiosis is or can be associated to several essential functions in AD or neurodegenerative disorders.

I enjoyed reading the review and learnt many issues related to Alzheimer’s disease and dysbiosis. The paper gave an impression for further studies and missing links in the AD pathogenesis, which must be reconsidered by many researchers.

Response: Thanks for comments
